# A novel pathogenic avipoxvirus infecting oriental turtle dove (*Streptopelia orientalis*) in China shows a high genomic and evolutionary proximity with the pigeon avipoxviruses isolated globally

Lei He,[1] Yuhao Zhang,[1] Yanyan Jia,[1] Zedian Li,[1] Jing Li,[1] Ke Shang,[1] Ke Ding,[1] Haotong Yu,[1] Subir Sarker[2]

**ABSTRACT**   Avipoxviruses are considered as significant viral pathogen infecting a wide range of domestic and wild bird species globally, yet the majority of avipoxviruses that infect the wild bird species remain uncharacterized and their genetic diversities remain unclear. In this study, we present a novel pathogenic avipoxvirus isolated from the cutaneous pox lesions of a wild oriental turtle dove (*Streptopelia orientalis*), tentatively named as turtle dovepox virus (TDPV). The avipoxvirus was isolated by using the chorioallantoic membranes of specific pathogen-free chicken embryos which showed characteristic focal pock lesions, followed by cytopathic effects in host cells infected with oriental turtle dovepox virus. An effort in sequencing the whole genome of the poxvirus using next-generation sequencing was given, and the first whole genome sequence of TDPV was obtained. The TDPV genome was 281,386 bp in length and contained 380 predicted open reading frames (ORFs). While 336 of the predicted ORFs showed homology to other characterized avipoxviruses, the other 44 ORFs were unique. Subsequent phylogenetic analyses showed that the novel TDPV shared the closest genetic evolutionary linkage with the avipoxviruses isolated from pigeon in South Africa and India, of which the TDPV genome had the highest sequence similarity (92.5%) with South African pigeonpox virus (FeP2). In conclusion, the sequenced TDPV is significantly different from any other avipoxviruses isolated from avian or other natural host species considering genomic architecture and observed sequence similarity index. Thus, it likely should be considered a separate species.

**IMPORTANCE**   Over the past few decades, avipoxviruses have been found in a number of wild bird species including the oriental turtle dove. However, there is no whole genome sequence information on avipoxviruses isolated from oriental turtle dove, leaving us unclear about the evolutionary linkage of avipoxviruses in oriental turtle dove and other wild bird species. Thus, we believe that our study makes a significant contribution because it is the first report of the whole genome sequence of TDPV isolated from a wild oriental turtle dove, which enriches the genomic information of the genus *Avipoxvirus*, furthermore, contributes to tracking the genetic evolution of avipoxviruses-infected oriental turtle dove species.

**KEYWORDS**   avipoxvirus, oriental turtle dove, next-generation sequencing, comparative genomics, phylogenetics, virus evolution

A vipoxviruses belong to the genus *Avipoxvirus* in the subfamily *Chordopoxvirinae* of the family *Poxviridae*, which are oval or brick enveloped DNA viruses replicating within the cytoplasm of infected cells. Avipoxviruses have a relatively large genome

Address correspondence to Lei He, helei4280546@163.com, or Subir Sarker, Subir.sarker@jcu.edu.au.

Lei He and Yuhao Zhang contributed equally to this article. Author order was determined alphabetically.

The authors declare no conflict of interest.

consisting of double-stranded DNA arranged in a linear configuration, with a length ranging from 188 to >360 kb. Avipoxviruses are considered as significant viral pathogen that causes infection in more than 374 avian species from 23 orders worldwide, including domestic and wild birds, and a number of additional bird species are likely to be susceptible (1–4). Wild birds contribute to the arrival of avipoxviruses to a new host via species exchange, habitat change, migration, and some other possible factors (5–7). Infection with avipoxviruses causes significant economic losses in the poultry industry, mainly in the forms of reduced growth rates, decreased egg production, and increased mortality (8). In infected birds, two distinct clinical manifestations of avipoxviruses infection are most frequently reported, identified as cutaneous and diphtheritic. The most common form of avipoxviruses infection develops proliferative wart-like lesions of various sizes, which are commonly restricted to unfeathered areas of the body, including the face, eyes, legs, feet, and beak. Thus, it is called "dry" pox. Although the cutaneous lesions may cause secondary bacterial and fungal infections that aggravate the bird's condition, full recovery is relatively rapid. The second form of avipoxviruses infection is less common and called "wet" or "diphtheritic" pox, which is characterized by proliferative lesions on the mucous membranes of the upper respiratory and alimentary tracts, which can cause asphyxia associated with a higher mortality rate (9–12).

Although avipoxviruses are capable of infecting a relatively large number of bird species worldwide, little information is available on the complete genomic characterization of avipoxviruses. According to the report published by the International Committee on Taxonomy of Viruses (ICTV), at present, only 12 species within the genus *Avipoxvirus* are registered, including *Canarypox virus, Flamingopox virus, Fowlpox virus, Juncopox virus, Mynahpox virus, Penguinpox virus, Pigeonpox virus, Psittacinepox virus, Quailpox virus, Sparrowpox virus, Starlingpox virus*, and *Turkeypox virus* (13). Nevertheless, presently, there are only limited quantities of available avipoxvirus whole genome sequences belonging to species recognized by ICTV in GenBank database, including 23 fowlpox viruses (14–19), two penguinpox viruses (20, 21), two pigeonpox viruses (20), a canarypox virus (22), a turkeypox virus (23), and a flamingopox virus (2). In addition, there are 10 further complete genomes of avipoxviruses available in GenBank including two shearwaterpox viruses (6), two magpiepox viruses (24, 25), two albatrosspox viruses (26, 27), a mudlarkpox virus (7), a cook's petrelpox virus (28), a crowpox virus (29), and a finch poxvirus (30), which are not among the species recognized by ICTV.

The oriental turtle dove (*Streptopelia orientalis*) belongs to the Order *Pigeoniformes*, family *Columbidae*. Currently, the oriental turtle dove is listed as "least concern" under the International Union for Conservation of Nature (IUCN) Red List of threatened species (31). Geographically, the existing breeding locations of oriental turtle dove are mainly distributed in China, India, Japan, Russian Federation (Central Asian Russia and Eastern Asian Russia), and some other countries around the Himalayas, Northeast Asia, and South Asia (32). Although the population trend of the oriental turtle dove is currently described as "stable" by the IUCN, it is essential to be aware of some potential factors affecting the population of the species, such as avipoxviruses infection that have been identified as a significant risk factor for the conservation of the threatened bird's population (5, 21, 33, 34).

Over the past few decades, although the avipoxviruses infection in many members of the family *Columbidae* has been reported worldwide (1, 3, 35), there have been only three reports of avipoxvirus infection in oriental turtle dove, which two reports occurring in Korea and one report occurred in China (36–38). To our knowledge, not one available complete genome sequence of avipoxvirus has been isolated and sequenced from oriental turtle dove infected with avipoxviruses. Therefore, this paper aims to identify and characterize a novel complete genome sequence of oriental turtle dovepox virus (TDPV) from an oriental turtle dove (*Streptopelia orientalis*) that was isolated from Henan, China, in 2021. Meanwhile, this is the first report of avipoxvirus complete genome sequence from the oriental turtle dove.

## MATERIALS AND METHODS

### Sampling

In August 2021, a sick oriental turtle dove was found in the campus of Henan University of Science and Technology (34°60′83.63″N, 112°42′84.59″E) in Luoyang City, Henan Province, China, and sent to the Key Lab of Animal Disease and Public Health (Henan University of Science and Technology) for molecular diagnosis. Nodular skin lesions were found on the toes, orbit, and wing root of the oriental turtle dove. The nodular skin lesion material was aseptically dissected and placed into a 2-mL microcentrifuge tube containing 1-mL sterile phosphate-buffered saline (PBS) with penicillin (200 U/mL) and streptomycin (200 µg/mL). After being kept at 4°C for 1 hour, to homogenize the material, Tissue Lyser (Servicebio, Wuhan, China) was used to grind the material with a vibration frequency of 25 times per second for 3 minutes, repeated three times. Thereafter, the suspensions were centrifuged at $5000 \times g$ for 15 minutes at 4°C, and the supernatant was stored at −20°C for subsequent virus isolation.

### Virus isolation

For avipoxvirus isolation, 100 µL of the supernatant of the suspensions was inoculated onto the chorioallantoic membranes (CAMs) of 10-day-old specific pathogen-free (SPF) chicken embryos obtained from the SPF Experimental Animal Center of Xinxing Dahua Agricultural, Poultry and Egg Co., Ltd., approved number SCXK (Guangdong) 2018-0019. The inoculated eggs were incubated at 37°C for 7 days and observed daily for mortality, followed by examination for the presence of focal white pock lesions or generalized thickening of the CAMs. The CAMs with pock lesions were harvested and similarly homogenized. Subsequently, the 100 µL supernatant of homogenized lysate was inoculated onto monolayers of Baby Hamster Syrian Kidney (BHK-21) cells and UMNSAH/DF-1 cells cultured in Dulbecco's Modified Eagle Medium (DMEM, Thermo-Fisher Scientific, USA) containing 10% fetal bovine serum, penicillin (100 U/mL), and streptomycin (100 µg/mL) at 37°C and 5% $CO_2$. After incubation for 2 hours, the cells were washed two times with sterile PBS, and then, DMEM that contained 2% fetal bovine serum, penicillin, and streptomycin was added. The cytopathic effects (CPEs) were observed at 96 hours post-inoculation.

### DNA extraction and sequencing

Viral DNA was extracted from the CAMs of SPF chicken embryos infected with avipoxvirus using a MiniBEST Viral RNA/DNA Extraction Kit (Takara, Dalian, China) according to the manufacturer's instructions. The next-generation sequencing and library construction were undertaken by Shanghai Tanpu Biotechnology Co., Ltd (Shanghai, China) to obtain primary sequence data. Briefly, the sequencing libraries were prepared using the TruSeq DNA Sample Prep Kit (Illumina, San Diego, CA, USA) as recommended by the manufacturer. PCR amplification of 10 cycles was performed after adapter ligation for sequencing target enrichment. The library was normalized and pooled in equimolar quantities, denatured, and diluted to optimal concentration before sequencing. The Illumina NovaSeq 6000 (Illumina, San Diego, CA, USA) was performed for sequencing to generate pair-end 150-bp reads.

### Assembly protocol

The resulting 36,543,760 paired raw sequence reads from NovaSeq 6000 were used to obtain the complete genome of TDPV. Initial quality control of all raw reads was generated, and the raw reads were processed by fastp (version 0.20.0 https://github.com/OpenGene/fastp) (39) for filtering to remove sequencing adapters, ambiguous bases, polyclonal reads, and poor-quality reads, including reads with read scores below Q20. Reads trimmed of adaptor sequences shorter than 50 nt were also discarded. The trimmed sequence reads are filtered by read-mapping using

the BBMap program (version 38.51 https://github.com/BioInfoTools/BBMap) (40) to remove likely contamination from host DNA, ribosomal RNAs, and bacteria. Unmapped reads were used as input data for *de novo* assembly using SPAdes (version 3.14.1 https://github.com/ablab/spades) (41) and SOAPdenovo (version 2.04 https://github.com/aquaskyline/SOAPdenovo-Trans) (42). Two large contigs were generated (275,343 bp and 2,801 bp), as well as several other small contigs corresponding to avipoxviruses sequences, according to searches of the GenBank database by BLASTp and BLASTn (https://blast.ncbi.nlm.nih.gov/Blast.cgi) (43). The overlaps and gaps between these contigs were further confirmed by site-specific PCR and Sanger sequencing to assemble the entire genome.

## Genome annotations

The assembled TDPV genome was initially annotated using the Genome Annotation Transfer Utility (44) with FeP2 genome (GenBank accession no. KJ801920.1) as the reference genome to predict all the potential open reading frames (ORFs), and the predicted ORFs were further verified using Geneious software (version 10.2.2 Biomatters, Auckland, New Zealand). ORFs longer than 30 amino acids with a methionine start codon (ATG) and not more than 50% overlap with other ORFs were used for selection and annotation. Subsequently, these ORFs were analyzed for similarities, including nucleotides (BLASTn) and proteins (BLASTx and BLASTp), which were considered for annotation as potential genes and numbered from left to right if they had significant sequence similarity to known viral or cellular genes (BLAST expect value $\leq e^{-5}$) or contained putative conserved domain that was predicted by BLASTp (45).

We followed the criteria for gene annotation process as described by Hendrickson et al. (46) and Carulei et al. (2). The ORFs were annotated as intact (I) if the 5′ end of the ORF is intact and the length of the ORF is ≥80% of the closest homolog. The ORFs were annotated as truncated (T)/fragment (F) if the length of the ORF is <80% of the closest homolog. The ORFs were annotated as extended (E) if both the 5′ and 3′ end of the ORF are intact or as extended at the 5′ and 3′ end, while the length of the ORF is >20% of the closest homolog. Furthermore, further examination of the final TDPV annotation with other poxvirus ortholog alignments was performed to determine the correct methionine start site, stop codons, and validity of overlaps.

In order to further understand the potential function of the unique ORFs tentatively identified in this study, multiple software was used to identify conserved domains or motifs as described by Sarker et al. (47). Transmembrane helices were predicted using the TMHMM server (https://services.healthtech.dtu.dk/services/TMHMM-2.0/) (48) and Geneious (version 10.2.2). Signal peptides were predicted using the SignalP (version 4.1) server (https://services.healthtech.dtu.dk/services/SignalP-4.1/) (49). Additionally, conserved domain was detected in the National Center for Biotechnology Information Conserved Domains Database (50). Tandem direct repeats were predicted using the Tandem Repeats Finders program (51).

## Comparative genomics

The genomic organization of the sequenced TDPV genome was visualized using Geneious software (version 2022.2.2). The determination of sequence similarity percentages between representative chordopoxvirus (ChPV) and TDPV complete genome sequences was performed using tools available in Geneious software (version 2022.2.2). Dot plots were built based on the EMBOSS dottup program available in Geneious software (version 2022.2.2), with word size of 12 (52).

## Phylogenetic analysis

Phylogenetic analysis of the novel TDPV genome sequence identified in this study was performed with other selected avipoxviruses genome sequences available in GenBank database. Genome sequences of each completely sequenced avipoxviruses (Table 1)

**TABLE 1** Comparative analysis of representative avipoxviruses and TDPV based on complete genome nucleotide sequences

| Avipoxviruses | Abbreviation | Genome identity (%) | Genome length (Kbp) | A + T content (%) | Number of ORFs | GenBank accession number | References |
|---|---|---|---|---|---|---|---|
| Oriental turtle dovepox virus | TDPV | | 281 | 70.4 | 380 | OQ547902 | |
| Albatrosspox virus | ALPV | 46.7 | 352 | 71.2 | 336 | MW365933 | (26) |
| Albatrosspox virus 2 | ALPV2 | 73.3 | 286 | 69.1 | 359 | OK348853 | (27) |
| Canarypox virus | CNPV | 45.4 | 360 | 69.6 | 328 | AY318871 | (22) |
| Cook's petrelpox virus | CPPV | 81.4 | 314 | 70.3 | 358 | OP292971 | (28) |
| Crowpox virus | CRPV | 49.0 | 329 | 71.3 | 403 | ON408417 | (29) |
| Finch poxvirus | FIPV | 45.7 | 354 | 69.9 | 334 | OM869483 | (30) |
| Flamingopox virus | FGPV | 86.2 | 293 | 70.5 | 285 | MF678796 | (2) |
| Fowlpox virus | FWPV | 72.6 | 289 | 69.1 | 260 | AF198100 | (55) |
| Magpiepox virus | MPPV | 51.8 | 293 | 70.4 | 301 | MK903864 | (24) |
| Magpiepox virus 2 | MPPV2 | 51.2 | 298 | 70.5 | 419 | MW485973 | (25) |
| Mudlarkpox virus | MLPV | 46.9 | 343 | 70.2 | 352 | MT978051 | (7) |
| Penguinpox virus | PEPV | 81.8 | 307 | 70.5 | 285 | KJ859677 | (20) |
| Penguinpox virus 2 | PEPV2 | 46.8 | 350 | 69.9 | 327 | MW296038 | (21) |
| Pigeonpox virus | FeP2 | 92.5 | 282 | 70.5 | 271 | KJ801920 | (20) |
| Pigeonpox virus | PPV | 91.8 | 280 | 70.5 | 252 | ON375849 | *[a] |
| Shearwaterpox virus 1 | SWPV1 | 49.5 | 327 | 72.4 | 310 | KX857216 | (6) |
| Shearwaterpox virus 2 | SWPV2 | 46.2 | 351 | 69.8 | 312 | KX857215 | (6) |
| Turkeypox virus | TKPV | 39.8 | 189 | 70.2 | 171 | KP728110 | (23) |

[a]Unpublished.

were obtained from GenBank database and were used for further phylogenetic analysis. Concatenated amino acid sequences of the selected nine poxvirus core proteins (NTPase, DNA polymerase, RNA polymerase subunit RPO147, RNA polymerase-associated protein PAP94, mRNA capping enzyme large subunit, virion core protein P4b, early transcription factor large subunit VETFL, virion core protein P4a, and RNA polymerase subunit RPO132) as well as the nucleotide sequences of DNA polymerase gene and P4b gene were aligned by MAFTT (version 7.490) with L-INS-I (gap open penalty 1.53; offset value 0.123) algorithm performed in Geneious software (version 2022.2.2) (53). Subsequently, maximum-likelihood (ML) analysis was performed in MEGA software (version 11.0.11) (54). Phylogenetic analysis for both nucleotide sequences of the P4b gene and DNA polymerase gene was performed based on the general-time-reversible model with gamma distribution rate variation and a proportion of invariable sites (GTR + G + I) with 1,000 bootstrap replicates in MEGA (version 11.0.11), while phylogenetic analysis for concatenated amino acid sequences was performed using the WAG model with 1,000 bootstrap replicates in MEGA (version 11.0.11).

## RESULTS

### Evidence of poxvirus infection in wild oriental turtle dove (*Streptopelia orientalis*)

The oriental turtle dove was clinically exhibiting depression, dullness, and anorexia. There were marked yellow nodular lesions observed in the eyes, with severe ulceration causing adhesion of the upper and lower eyelids, followed by obvious yellow nodular lesions on paws, and pox scabs left at the wing roots after the pox-like lesions had fallen off accompanied by skin ulceration (Fig. 1A through C).

To investigate the presence of viral pathogen in the sick oriental turtle dove, the supernatant extracted from the cutaneous pox lesions was inoculated into the CAMs of SPF chicken embryos, which showed a characteristic focal pale pock lesions with moderate thickening and swelling of CAMs (Fig. 1E). Besides, typical CPE was observed in both BHK-21 cells and DF-1 cells after 96 hours of inoculation with oriental turtle dovepox virus isolated in this study. The infected BHK-21 cells exhibited cell rounding,

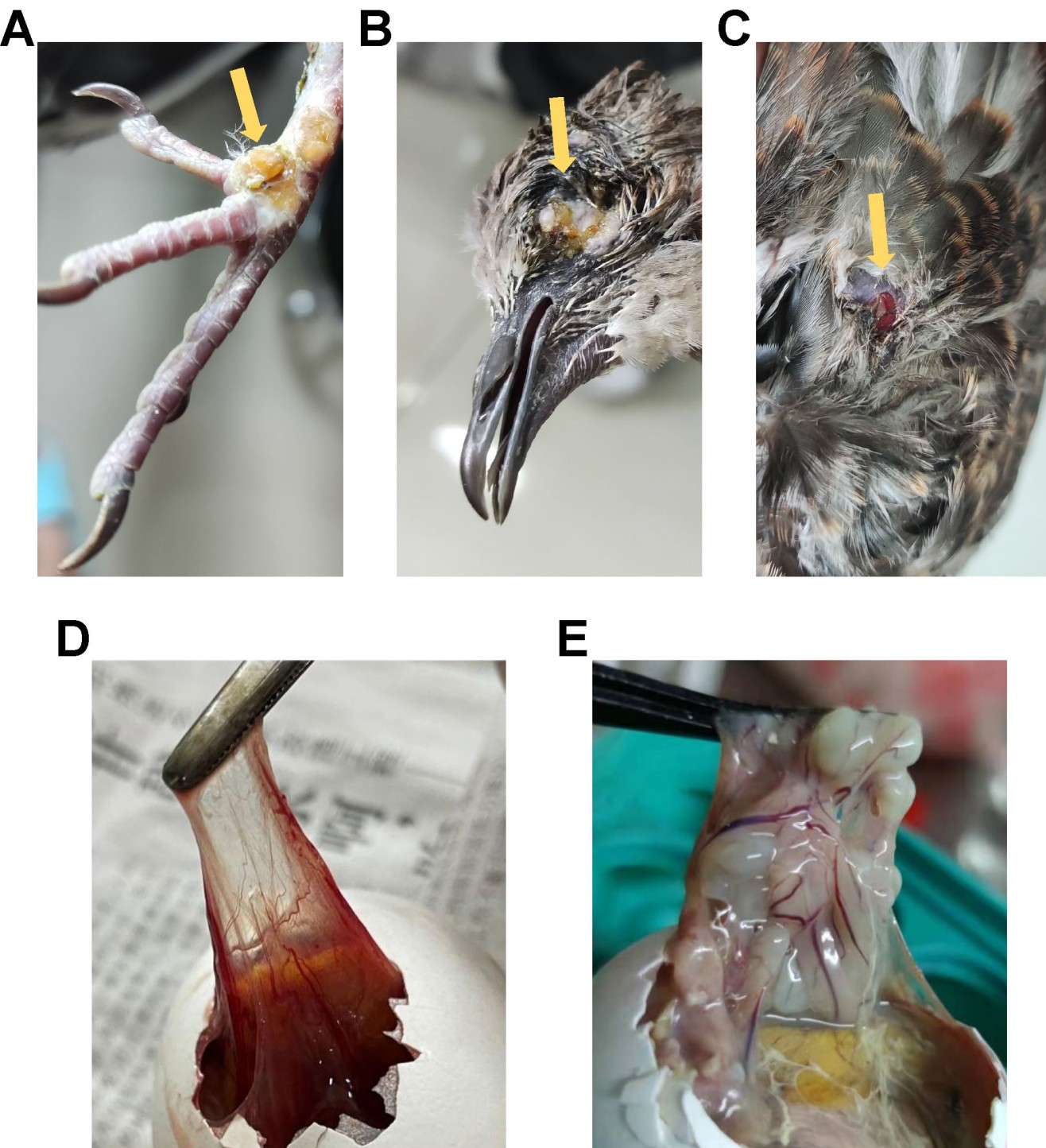

**FIG 1** Gross lesion pictures and poxvirus-specific lesions in the CAMs of SPF chicken embryos induced by TDPV. As shown by the yellow arrow, the marked yellow nodular lesions were observed on the paws (A) and eyes (B), and the pox scabs left at the wing roots after the pox-like lesions had fallen off accompanied by skin ulceration (C). (D) Control group: uninoculated CAMs of SPF chicken embryos. (E) Experiment group: the CAMs of 10-day-old SPF chicken embryos inoculated with the lesion lysate supernatant and harvested at 7 days post-inoculation, and the characteristic focal pale pock lesions with moderate thickening and swelling of CAMs from inoculated chicken embryos can be observed.

severe aggregation, and massive detachment with empty plaques compared to the normal cells (Fig. 2A and B). Likewise, compared with uninfected DF-1 cells, most cells appeared rounded, aggregated, and partially detached in infected DF-1 cells (Fig. 2C and

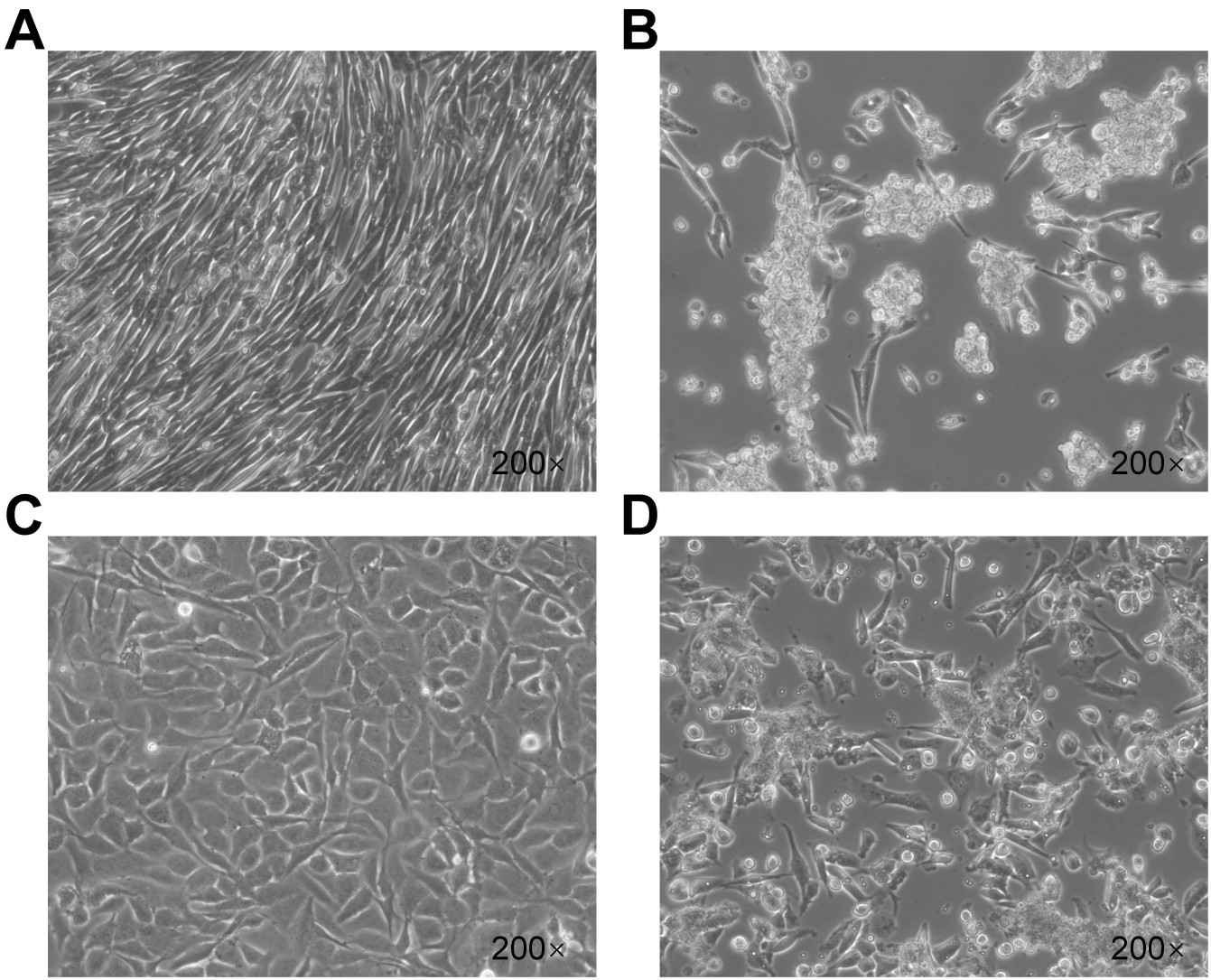

**FIG 2** Typical CPE was observed in BHK-21 cells and DF-1 cells induced by TDPV at 96 hours post-infection. Control group: Uninoculated BHK-21 cells (A) and DF-1 cells (C). Experiment group: BHK-21 cells (B) and DF-1 cells (D) infected with the homogenized supernatants of CAMs with pock lesions.

D). Meanwhile, poxvirus infection was further confirmed by performing a PCR targeting approximately 578 bp in length of P4b gene, of which resultant the existence of avipoxvirus infection in oriental turtle dove (data not shown).

## Genome structure of oriental turtle dovepox virus (TDPV)

The TDPV complete genome was assembled into a contiguous sequence of linear double-stranded DNA molecule of 281,386 bp in length and submitted to the GenBank under the accession number OQ547902. The genome of TDPV encompassed a well-conserved central coding region bounded by two matching inverted terminal repeat (ITR) regions, constituting 3,116 bp each (coordinates 1–3,116 sense and 278,271–281,386 antisense orientation) similar to most of other avipoxviruses (6, 7, 21, 24, 26, 27). Each of the ITR regions composed arrays of direct repeats, and three tandem repeats were detected within each ITR region, which were consisted of a 56-bp, 90-bp, and 34-bp repeat unit and shared approximately 98–100% nucleotide identity. The A + T content of the TDPV complete genome was detected to be 70.4%, which corresponds very well to other avipoxviruses isolated from other avian host species such as Australian magpie (24, 25), pigeon (20), lesser flamingos (2), African penguin (20), and Cook's petrel (28) (Table

1). The TDPV complete genome showed the highest nucleotide identity (92.5%) with the pathogenic pigeonpox virus isolated from a feral pigeon (*Columba livia*) in South Africa in 2004 (GenBank accession no. KJ801920) (20), followed by PPV (91.8%), FGPV (86.2%), PEPV (81.8%), CPPV (81.4%), ALPV2 (73.3%), and FWPV (72.6%) (Table 1).

## Genome annotation and comparative analyses of TDPV

The TDPV genome was predicted to contain 380 methionine-initiated ORFs encoding proteins within the range of 30 to 1,923 amino acids in length, which were annotated as putative genes and numbered from left to right (Fig. 3; Table S1). Among them, seven predicted ORFs of TDPV genome are located within ITR regions, therefore, exist as duplex copies, which correspond to ORF001-007 and ORF374-380. After the comparative analysis of the protein sequences encoded by the predicted ORFs, we found a large number of ORFs (336) shared the greatest similarity with the gene products of other ChPVs (E value ≤$10^{-5}$) (Fig. 3; Table S1). Among these predicted ORFs of TDPV, the largest number of genes (182) was demonstrated the highest similarity to ORFs of pigeonpox virus (FeP2) isolated from a feral pigeon in South Africa (20). A further 58 genes showed the highest similarity to CPPV, followed by 51 genes to FGPV, 31 genes to PEPV, nine genes (ORF028, -30, -49, -92, -126, -155, -165, -182, and -301) to PPV, and five genes (ORF012, -112, -151, -195, and -351) to FWPV (Fig. 3; Table S1). Remarkably, all the conserved genes predicted in the TDPV genome have the highest similarity to FeP2, and these observations indicate that the conserved genes of TDPV share a common

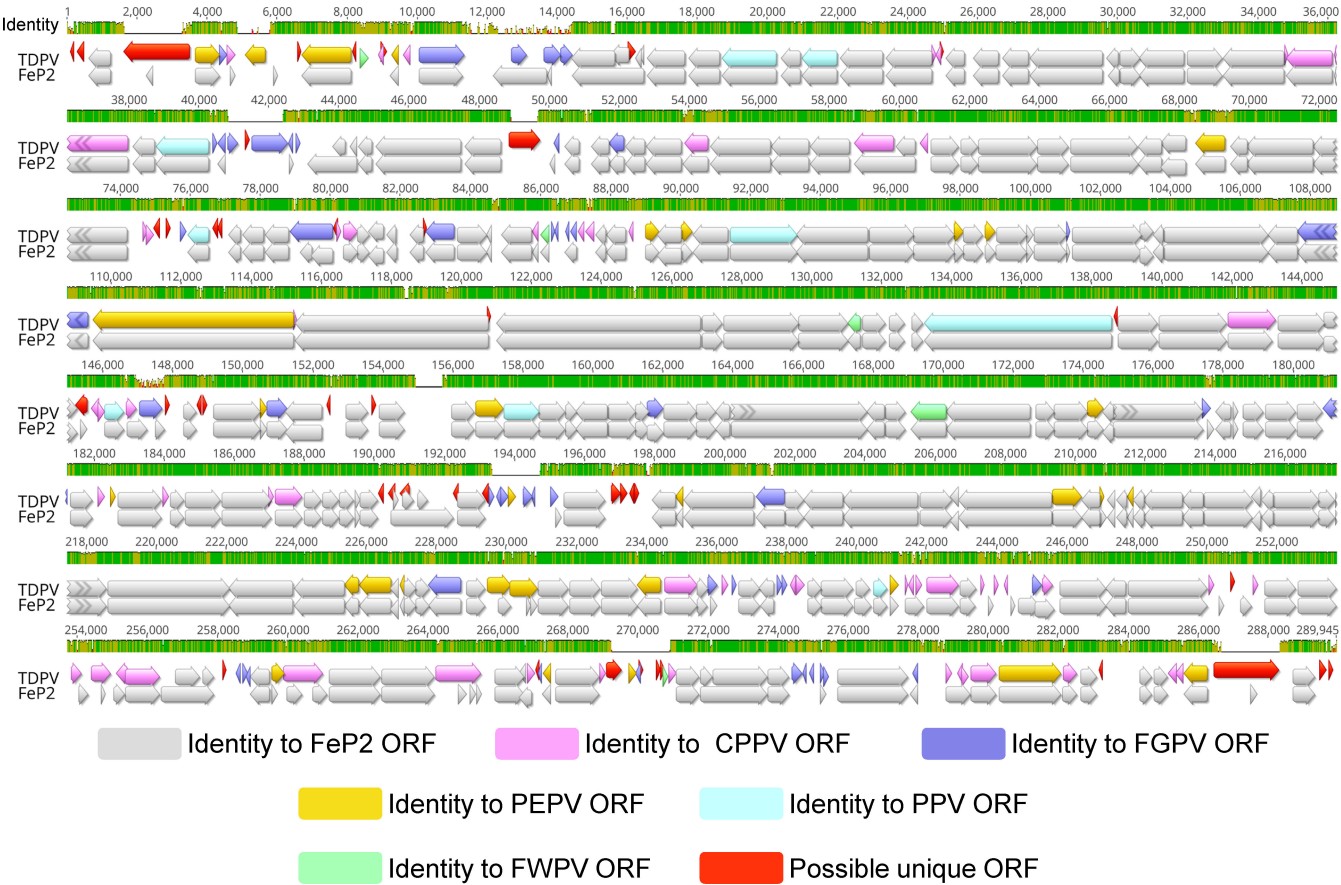

**FIG 3** The comparison illustration between the genome of TDPV and FeP2. A sequence alignment was performed using MAFFT in geneious prime software (version 2022.2.2) to compare the ORFs map between TDPV and FeP2. The arrows represent the transcriptional direction of the genes and ORFs, and each gene or ORF in TDPV was assigned a corresponding color based on other avipoxvirus that was homologous to it, as indicated by the key in the legend. The top graph represents the mean pairwise sequence identity over all pairs in the column between TDPV and FeP2 (green: 100% identity; mustard: ≥30% and <100% identity; red: <30% identity).

evolutionary history with the poxviruses that infect pigeon species. In comparison of the predicted ORFs in TDPV with the most similar homologs in other poxvirus genomes, 239 ORFs have been annotated as intact; 94 ORFs, as truncated and/or fragmented; and three ORFs, as extended (Table S1).

Interestingly, the TDPV genome contained 44 predicted protein-coding genes, including six genes (ORF001, -002, -004, -377, -379, and -380) located within the ITR regions that were not predicted within any other poxvirus genome, neither matched to any sequences in the NR protein database using BLASTX and BLASTP in NCBI. These unique ORFs encoded viral proteins in the length range of 30–137 amino acids. Furthermore, only one unique protein coding gene (ORF-011) was predicted to contain a single transmembrane helix (TMH) using the TMHMM server (Table S1).

The multiple genome alignment based on nucleotide sequences was performed using MAFFT software; the TDPV was shown to share 92.5% similarity compared to FeP2, followed by the 91.8%, 86.2%, 81.8%, and 81.4% identity to PPV, FGPV, PEPV, and CPPV, respectively (Table 1). On the other hand, to better understand the overall genomic synteny of TDPV with other selected avipoxviruses, we used the dot plot analysis. The results of the dot plot analyses were similar to those of the multiple genome alignment. The TDPV genome was highly syntenic with FeP2, PPV, and FGPV (Fig. S1A through C), and the TDPV genome also displayed significant differences compared to the whole genome of FIPV, CNPV, and TKPV (Fig. S1D through F).

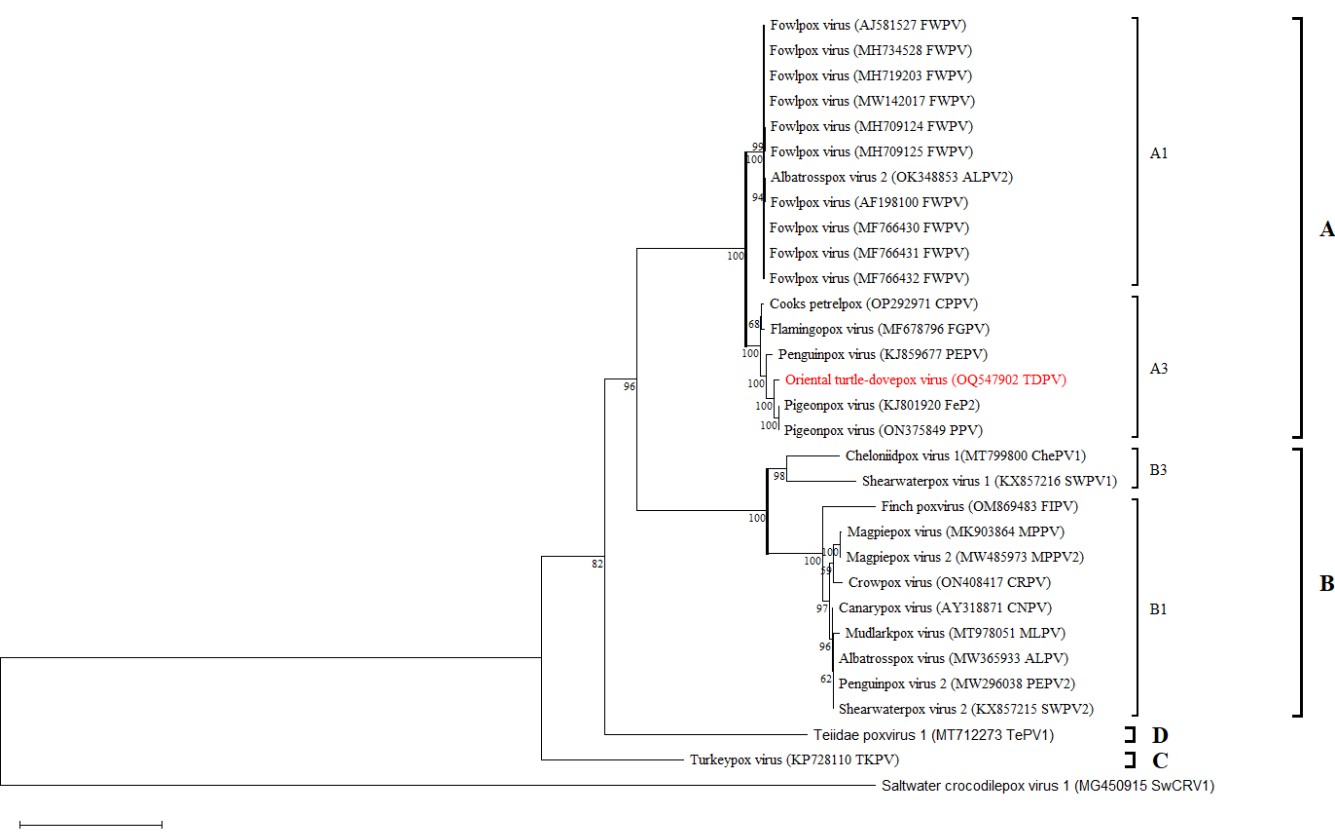

**FIG 4** Phylogenetic relationships between oriental TDPV and other chordopoxviruses. An ML tree was constructed from multiple alignments of the concatenated amino acid sequences of the selected nine poxvirus core proteins using MEGA software (version 11.0.11) with 1,000 bootstraps. The numbers on the left show bootstrap values as percentages (0–100). The labels at branch tips refer to virus species, followed by GenBank accession numbers and abbreviated species names in parentheses. Details of the poxviruses used in the phylogenetic tree can be seen in Table 2. Saltwater crocodile poxvirus 1 was selected as an outgroup. The position of novel TDPV is highlighted using red text, and the major clades and sub-clades are designated according to Gyuranecz et al. (3).

## Evolutionary relationships of TDPV

The ML tree of the concatenated amino acid sequences provides a clear indication of the fact that TDPV belongs to the genus *Avipoxvirus*. In the ML tree (Fig. 4), TDPV was located within a subclade A3 encompassing avipoxviruses isolated from feral pigeon (*Columba livia*), pigeon, African penguin (*Spheniscus demersus*), and lesser flamingos (*Phoenicopterus minor*) with robust bootstrap support (100%), indicating that it may be representing an ancient evolutionary lineage within the genus, *Avipoxvirus*. In subclade A3, PEPV is the root of the TDPV, FeP2, and PPV, implying that these avipoxviruses under this subclade likely evolved from African penguin. Thus, the FeP2 and PPV as well as the newly isolated TDPV have all evolved from the PEPV. Using the same set of concatenated ChPV core protein sequences, the maximum inter-lineage sequence identity values were calculated to be between 98.3% and 99.2% of the four avipoxviruses (FeP2, PEPV, CPPV, and FGPV) under the subclade A3 of the ML tree (Fig. S2). Among them, the sequence identity values were 99.2% (TDPV vs FeP2), 99.2% (TDPV vs PPV), 98.5% (TDPV vs PEPV), 98.4% (TDPV vs CPPV), and 98.3% (TDPV vs FGPV), which mirrored the phylogenetic position of this newly avipoxvirus originated from an oriental turtle dove, and it can be further inferred that these avipoxviruses most likely evolved from a likely common ancestor. Furthermore, phylogenetic analysis using partial nucleotide sequences of the P4b gene (Fig. S3) and DNA polymerase gene (Fig. S4) likewise demonstrated the evolutionary genetic linkage of some other avipoxviruses with TDPV in this study. Among them, we discovered that the avipoxviruses isolated from oriental turtle doves in China (37) and South Korea (3) and a great bustard in Spain (3) were most identical to the newly isolated TDPV within a correspondingly small fragment of the genome.

## DISCUSSION

This paper reports an identification and characterization of the first whole genome of a novel avipoxvirus (TDPV) from an infected oriental turtle dove in China. The first description of avipoxvirus infection in oriental turtle dove was reported in 2011 and evidenced by PCR to amplify the P4b gene of the avipoxvirus (36). Currently, no taxonomic classification has been granted for TDPV by the ICTV (https://talk.ictvon-line.org/taxonomy/), and no evolutionary relationship has been established with other members of the family *Poxviridae* due to the lack of the available data about whole genome sequence of TDPV (13).

In this study, the whole genome sequence of TDPV was determined by using the next-generation sequencing on the samples collected from the naturally occurring pox lesions on the skin of infected oriental turtle dove and performing virus isolation using the CAMs of SPF chicken embryos. In addition, typical CPEs were observed in both isolated TDPV-infected BHK-21 cells and DF-1 cells. Overall, the results demonstrated that the structure of the TDPV genome, including genome size, A + T content, and number of ORFs, was remarkably consistent with other known avipoxviruses in the GenBank database. Nevertheless, the TDPV genome was also significantly different from those other avipoxviruses in some extent, where the whole genome sequence of TDPV showed substantial distinctive similarity to the other avipoxviruses, but had the highest sequence similarity to FeP2 (92.5%) (20), PPV (91.8%), FGPV (86.2%) (2), PEPV (81.8%) (20), and CPPV (81.4%) (28) (Table 1), as well as the most matching syntenic with FeP2 (Fig. S1A). Furthermore, the TDPV genome was missing five genes at the corresponding positions compared to the most similar FeP2, and the 44 predicted protein-coding genes within its genome cannot be found in any other poxvirus (Table S1). Meanwhile, there were some truncated or fragment ORFs of this TDPV genome, which remain to be further investigated as to whether they have the corresponding function. In general, the TDPV was genetically different sufficiently from other avipoxviruses that it could be considered a new virus species within the genus *Avipoxvirus*.

As shown in Fig. 4, the phylogenetic tree constructed using the concatenated amino acid sequences of the ChPV core gene provides evidence that TDPV was most closely related to FeP2, PPV, PEPV, FGPV, and CPPV, suggesting that these avipoxviruses likely

**TABLE 2** Related poxvirus genome sequences used in further analysis of TDPV

| Poxviruses | Abbreviation | Year of isolation[a] | Country of origin[a] | GenBank accession number | References |
|---|---|---|---|---|---|
| Oriental turtle dovepox virus | TDPV | 2021 | China | OQ547902 | |
| Albatrosspox virus | ALPV | 1997 | New Zealand | MW365933 | (26) |
| Albatrosspox virus 2 | ALPV2 | 1997 | New Zealand | OK348853 | (27) |
| Canarypox virus | CNPV | 1948 | USA | AY318871 | (22) |
| Cheloniidpox virus 1 | ChePV1 | 2018 | Australia | MT799800 | (47) |
| Cook's petrelpox virus | CPPV | 2022 | Australia | OP292971 | (28) |
| Crowpox virus | CRPV | 2021 | Australia | ON408417 | (29) |
| Finch poxvirus | FIPV | 2021 | USA | OM869483 | (30) |
| Flamingopox virus | FGPV | 2008 | South Africa | MF678796 | (2) |
| Fowlpox virus | FWPV | 1997 | Australia | MW142017 | (16–18, 55) |
| | | 1999 | USA | AF198100 | |
| | | 2003 | UK | AJ581527 | |
| | | 2015 | USA | MH734528 | |
| | | 2015 | USA | MH719203 | |
| | | 2016 | France | MF766430-32 | |
| | | 2018 | USA | MH709124-25 | |
| Magpiepox virus | MPPV | 2018 | Australia | MK903864 | (24) |
| Magpiepox virus 2 | MPPV2 | 1956 | Australia | MW485973 | (25) |
| Mudlarkpox virus | MLPV | 2019 | Australia | MT978051 | (7) |
| Penguinpox virus | PEPV | 2014 | South Africa | KJ859677 | (20) |
| Penguinpox virus 2 | PEPV2 | 1997 | New Zealand | MW296038 | (21) |
| Pigeonpox virus | FeP2 | 2014 | South Africa | KJ801920 | (20) |
| Pigeonpox virus | PPV | 2022 | India | ON375849 | *[b] |
| Saltwater crocodilepox virus 1 | SwCRV1 | 2017 | Australia | MG450915 | (56) |
| Shearwaterpox virus 1 | SWPV1 | 2015 | Australia | KX857216 | (6) |
| Shearwaterpox virus 2 | SWPV2 | 2015 | Australia | KX857215 | (6) |
| Turkeypox virus | TKPV | 2011 | Hungary | KP728110 | (23) |
| Teiidae poxvirus 1 | TePV1 | 2019 | Australia | MT712273 | (57) |

[a]If the collection date/country was not available, the year/country of submission to GenBank is reported.
[b]Unpublished.

originated from a common ancestor. Meanwhile, based on the ML tree, we can postulate that the FeP2 and PPV isolated in 2014 and 2023, respectively (20), are likely to originate from a common ancestor as well as TDPV isolated in this study. A well-supported ML phylogenetic tree constructed using the nucleotide sequences of the P4b gene and DNA polymerase gene of the avipoxvirus genome revealed that the TDPV was part of the subclade A3, which also contained FeP2, PPV, FGPV, PEPV, CPPV, and other avipoxviruses isolated around the world (Fig. S3 and S4) (2, 3, 20). These results further evidence the close linkage of TDPV to avipoxviruses isolated from diverse avian species at the level of a conserved gene. Remarkably, among the avipoxviruses within subclade A3, which were most closely linked to the genetic evolution of TDPV, some avipoxviruses were isolated from oriental turtle dove, feral pigeon, and rock dove, all belonging to the family *Columbidae* (3, 20, 37). However, an exception was an avipoxvirus isolated from a great bustard, which belongs to the *Otididae*, which may imply a possibility of cross-species transmission of avipoxviruses which can infect the species of family *Columbidae*, switch to infect species belonging to the family *Otididae*. Interestingly, as shown in Fig. S3, TDPV was most genetically related to a strain of avipoxvirus also isolated from an oriental turtle dove in China. According to Yuan et al. (37), this was a wild oriental turtle dove with a pox rash on its body found on the campus of Shandong University in 2020, and further viral isolation and sequence analysis of the P4b gene were performed. Regrettably, they did not perform further sequencing and analysis of the whole genome of the isolated avipoxvirus, which resulted in the whole genome information and phylogenetic relationship of this strain of avipoxvirus not being clearly clarified, so we could not

exclude the possibility that TDPV and the avipoxvirus belonged to the same virus at present.

The TDPV isolated in this study belongs to subclade A3 in terms of genetic classification, and most of the avipoxviruses within this subclade were isolated from infected species of the order *Columbiformes*, family *Columbidae* (3). Moreover, members of the family *Columbidae* are distributed worldwide as one of the largest avian host groups infected with avipoxviruses (1, 38). Although the first description of avipoxvirus infection in oriental turtle dove was in South Korea in 2011 (36), the report of avipoxvirus infection in pigeon with members of the family *Columbidae* can be traced back to 1849 in Germany (1). In the last few decades, cases of avipoxviruses infection in the species belonging to the family *Columbidae* have been reported frequently (1, 3, 4, 36, 58, 59). However, due to limited information, currently, the epidemiology, pathogenicity, and transmission ability of avipoxviruses infecting the species of family *Columbidae* are not clearly characterized. Thus, the series of characterization of avipoxviruses in the species of family *Columbidae* still necessitates further experimental demonstration. Some studies have shown that infected avian species are the main hosts of avipoxviruses and the most common route of transmission is through the bite of insects, mainly members of the midges (*Ceratopogonidae*) and mosquitos (*Culicidae*) thought to play a significant role in the mechanical transmission of avipoxviruses in wild bird populations (58, 60). Therefore, as with other avipoxviruses, the transmission of avipoxviruses in infected oriental turtle dove may also be mediated by insect vectors.

## Conclusions

This paper reports the discovery and whole genome characterization of a novel avipoxvirus isolated from oriental turtle dove in Henan Province, China, and tentatively named as oriental turtle dovepox virus (TDPV), under the genus *Avipoxvirus*. We found that the whole genome sequence of TDPV was significantly different from other known avipoxviruses by comparing the genome sequence similarity and genetic composition. Therefore, TDPV should be considered as a new species under the family *Poxviridae*, genus *Avipoxvirus*. The identification and characterization of the newly TDPV in this study enriched the genomic information of the genus avipoxvirus and enhanced our understanding of the species of avipoxvirus infecting family *Columbidae*, which also contributed to tracking the evolutionary linkage of avipoxvirus infecting this species. Nevertheless, further studies on the pathogenicity, epidemiology, cross-species transmission ability, and host specificity of TDPV infected the oriental turtle dove species, followed by obtaining and sequencing more whole genome sequences of avipoxviruses remain crucial.

### ACKNOWLEDGMENTS

This research was supported by the science and technology research project of Henan province (nos. 222102110012 and 232102110078) funded by the Chinese Government. S.S. is the recipient of an Australian Research Council Discovery Early Career Researcher Award (grant number DE200100367) funded by the Australian Government. The Chinese and Australian Government had no role in the study design, data collecting, data analysis, decision to publish, or preparation of the manuscript.

L.H., Y.Z., and S.S. conceptualized the study. Y.J., J.L., and Z.L. conducted the formal analysis; K.D., H.Y., and K.S. conducted the investigation; L.H. and Y.Z. wrote the original draft. S.S. revised and edited the manuscript. Funding was acquired by L.H.

All authors read and approved the final manuscript.

The authors declare that they have no conflicts of interest.

## AUTHOR AFFILIATIONS

[1]The Key Lab of Animal Disease and Public Health /Luoyang Key Laboratory of Live Carrier Biomaterial and Animal Disease Prevention and Control, Henan University of Science and Technology, Luoyang, Henan, China

[2]Biomedical Sciences & Molecular Biology, College of Public Health, Medical and Veterinary Sciences, James Cook University, Townsville, Australia

## AUTHOR ORCIDs

Yuhao Zhang http://orcid.org/0009-0000-9497-9819
Subir Sarker http://orcid.org/0000-0002-2685-8377

## AUTHOR CONTRIBUTIONS

Lei He, Conceptualization, Writing – original draft | Yuhao Zhang, Conceptualization, Writing – original draft | Yanyan Jia, Formal analysis | Zedian Li, Formal analysis | Jing Li, Formal analysis | Ke Shang, Investigation | Ke Ding, Investigation | Haotong Yu, Investigation | Subir Sarker, Conceptualization, Writing – review and editing

## DATA AVAILABILITY

The TDPV complete genome sequence and the associated data sets that were generated during this study were deposited in the GenBank database under the accession number OQ547902.

## ADDITIONAL FILES

The following material is available online.

### Supplemental Material

**Supplemental material (Spectrum01193-23-s0001.pdf).** Fig. S1 to S4; Table S1.

### Open Peer Review

**PEER REVIEW HISTORY (review-history.pdf).** An accounting of the reviewer comments and feedback.

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
