## [Reviewer comments · Microbiology Spectrum]

Microbiology Spectrum

A Novel Pathogenic Avipoxvirus Infecting Oriental Turtle Dove (*Streptopelia orientalis*) in China Shows a High Genomic and Evolutionary Proximity with the Pigeon Avipoxviruses Isolated Globally

Lei He, Yu Zhang, Yan Jia, Ze Li, Jing Li, Ke Shang, Ke Ding, Hao Yu, and Subir Sarker

Corresponding Author(s): Lei He, Henan University of Science and Technology

Review Timeline:

Submission Date:	March 18, 2023
Editorial Decision:	May 7, 2023
Revision Received:	May 18, 2023
Editorial Decision:	May 24, 2023
Revision Received:	May 31, 2023
Editorial Decision:	June 23, 2023
Revision Received:	August 7, 2023
Accepted:	August 8, 2023

Editor: Wen Chang

Reviewer(s): The reviewers have opted to remain anonymous.

Transaction Report:

DOI: <https://doi.org/10.1128/spectrum.01193-23>

May 7, 2023

Dr. Lei He
Henan University of Science and Technology
Luoyang
China

Re: Spectrum01193-23 (A Novel Pathogenic Avipoxvirus Infecting Oriental Turtle Dove (*Streptopelia orientalis*) in China Shows a High Genomic and Evolutionary Proximity with the Pigeon Avipoxviruses Isolated Globally)

Dear Dr. Lei He:

Thank you for submitting the manuscript to Microbiology Spectrum for publication. The manuscript is well written and the data support the conclusion. Before accepting the manuscript for publication, I would like to call your attention on Figure 3. The EM image of TDPV particles appeared around 100nm, according to the scale bar. It is too small for poxviruses including avian poxvirus. The negative staining needs to be repeated to show a representative staining of TDPV particles.

Link Not Available

Sincerely,

Wen Chang

Journals Department
Staff Comments:

Preparing Revision Guidelines

- Point-by-point responses to the issues raised by the reviewers in a file named "Response to Reviewers," NOT IN YOUR

COVER LETTER.

- Upload a compare copy of the manuscript (without figures) as a "Marked-Up Manuscript" file.
- Each figure must be uploaded as a separate file, and any multipanel figures must be assembled into one file.
- Manuscript: A .DOC version of the revised manuscript
- Figures: Editable, high-resolution, individual figure files are required at revision, TIFF or EPS files are preferred

Please return the manuscript within 60 days; if you cannot complete the modification within this time period, please contact me. If you do not wish to modify the manuscript and prefer to submit it to another journal, please notify me of your decision immediately so that the manuscript may be formally withdrawn from consideration by Microbiology Spectrum.

Editor/Reviewers

Comments and Suggestions for Authors

Thank you for submitting the manuscript to Microbiology Spectrum for publication. The manuscript is well written, and the data support the conclusion. Before accepting the manuscript for publication, I would like to call your attention on Figure 3. The EM image of TDPV particles appeared around 100nm, according to the scale bar. It is too small for poxviruses including avian poxvirus. The negative staining needs to be repeated to show a representative staining of TDPV particles.

Response: We are thankful for the efforts and suggestion made by the editor and reviewers. We believe that the revised manuscript has significantly improved. In response to editor suggestion, we have added below text and modified Figure 3.

“Moreover, transmission electron microscopic analysis of the supernatant samples from the infected CAMs of SPF chicken embryos and the cutaneous pox-lesions were carried out. Both the presence of immature virions (Fig. 3 A-B) and mature virions (Fig. 3 C-D) with ovoid morphology of virus particles and a visible outer envelope were observed. Besides, it is interesting to notice that the size of the viral particles varies with different forms measuring approximately 95×80 nm in diameter of immature virions (Fig A-B) and the mature virions were 360×415 nm in diameter (Fig C-D), indicating the presence of TDPV particles in various forms from the CAMs of SPF chicken embryos and the pox-lesions from wild oriental turtle dove”.

FIG 3 Transmission electron microscopic analysis of negatively stained sample sourced from the CAMs lysates of SPF chicken embryos infected with TDPV (Fig A-B) and the cutaneous pox-lesions of a wild oriental turtle dove (Fig C-D). Oriental turtle dovepox virus particles were shown by transmission electron microscopy, measuring approximately 95×80 nm in diameter of immature virions (Fig B, yellow arrow) and the mature virions were 360×415 nm in diameter (Fig D, red arrow).

May 24, 2023

Dr. Lei He
Henan University of Science and Technology
Luoyang
China

Re: Spectrum01193-23R1 (A Novel Pathogenic Avipoxvirus Infecting Oriental Turtle Dove (*Streptopelia orientalis*) in China Shows a High Genomic and Evolutionary Proximity with the Pigeon Avipoxviruses Isolated Globally)

Dear Dr. Lei He:

In the revised manuscript, the EM images of TDPV particles remained questionable when compared with other avian poxviruses in literatures (*Avian Pathology* 38(6):485-9 (2009).) One could also check the EM images of fowlpox, pigeon pox, quail pox and sparrow pox virus particles in the official NARO website (<https://www.naro.go.jp/english/laboratory/niah/em/virus/042926.html>). Since the viral DNA sequencing and bioinformatic analyses stand on their own, I suggest to remove Figure 3 in the final version to avoid confusion.

Thank you for submitting your manuscript to *Microbiology Spectrum*. As you will see your paper is very close to acceptance. Please modify the manuscript along the lines I have recommended. As these revisions are quite minor, I expect that you should be able to turn in the revised paper in less than 30 days, if not sooner. If your manuscript was reviewed, you will find the reviewers' comments below.

When submitting the revised version of your paper, please provide (1) point-by-point responses to the issues raised by the reviewers as file type "Response to Reviewers," not in your cover letter, and (2) a PDF file that indicates the changes from the original submission (by highlighting or underlining the changes) as file type "Marked Up Manuscript - For Review Only". Please use this link to submit your revised manuscript. Detailed instructions on submitting your revised paper are below.

Link Not Available

Thank you for the privilege of reviewing your work. Below you will find instructions from the *Microbiology Spectrum* editorial office and comments generated during the review.

Sincerely,

Wen Chang

Editor, *Microbiology Spectrum*

Reviewer comments:

Preparing Revision Guidelines

Please return the manuscript within 60 days; if you cannot complete the modification within this time period, please contact me. If you do not wish to modify the manuscript and prefer to submit it to another journal, please notify me of your decision immediately so that the manuscript may be formally withdrawn from consideration by Microbiology Spectrum.

Editor/Reviewers

Comments and Suggestions for Authors

In the revised manuscript, the EM images of TDPV particles remained questionable when compared with other avian poxviruses in literatures (*Avian Pathology* 38(6):485-9 (2009).) One could also check the EM images of fowlpox, pigeon pox, quail pox and sparrow pox virus particles in the official NARO website (<https://www.naro.go.jp/english/laboratory/niah/em/virus/042926.html>). Since the viral DNA sequencing and bioinformatic analyses stand on their own, I suggest to remove Figure 3 in the final version to avoid confusion.

Response: We are thankful for the efforts and suggestion made by the editor and reviewers. In the revised version, the Figure 3 and other statements about the EM images of TDPV has been removed. Besides, we have revised the manuscript carefully and some other drawbacks have been corrected and marked red. We believe that the revised manuscript has significantly improved. In response to editor suggestion, we have deleted the part about TEM images of TDPV particles.

June 21, 2023

Dr. Lei He
Henan University of Science and Technology
Luoyang
China

Re: Spectrum01193-23R2 (A Novel Pathogenic Avipoxvirus Infecting Oriental Turtle Dove (*Streptopelia orientalis*) in China Shows a High Genomic and Evolutionary Proximity with the Pigeon Avipoxviruses Isolated Globally)

Dear Dr. Lei He:

Link Not Available

Sincerely,

Wen Chang

Journals Department
Reviewer comments:

The authors present the isolation of a novel pathogenic avipoxvirus - designated turtledove pox virus (TDPV) - from *Streptopelia orientalis*. The virus was passaged through chorioallantoic membranes (CAMs) of 10-day-old specific 153 pathogen-free (SPF) chicken embryos and subsequently through BHK-21 cells.

Suitable evidence of the pathology in the bird is presented in figure 1. However, the cytopathic impact shown in figure 2 appears marginal in panel B. More cells appear to be rounded, but this figure does not clear evidence of plaques. Could another field of view be considered or clear labelling of the differences?

Electron microscopy was undertaken, and a virus-like structure identified, however this is not of high resolution and could be omitted (Figure 3).

The genome of the virus was determined through genome sequencing. Analysis of the differences with another pigeon virus revealed that this virus contains 44 unique open reading frames (Figure 4).

The phylogenetic relationships between the oriental turtle dove virus (TDPV) and 63 other chordopoxviruses were undertaken and place this novel virus within the family, suggesting evolutionary relations.

Staff Comments:

Preparing Revision Guidelines

Please return the manuscript within 60 days; if you cannot complete the modification within this time period, please contact me. If you do not wish to modify the manuscript and prefer to submit it to another journal, please notify me of your decision immediately so that the manuscript may be formally withdrawn from consideration by Microbiology Spectrum.

Editor/Reviewers

Comments and Suggestions for Authors

1. Comment:

The authors present the isolation of a novel pathogenic avipoxvirus-designated turtledove pox virus (TDPV)-from *Streptopelia orientalis*. The virus was passaged through chorioallantoic membranes (CAMs) of 10-day-old specific pathogen-free (SPF) chicken embryos and subsequently through BHK-21 cells.

Response:

Thank you for your comments and summary, we isolated TDPV from the cutaneous pox-lesions of a wild oriental turtle dove, and the TDPV was purified from CAMs of SPF chicken embryos. Meanwhile, we could observe the characteristic focal pale pock lesions with moderate thickening and swelling of CAMs. Thereafter, and the typical CPE (cell rounding, aggregation and detachment with empty plaques) can be observed of the BHK-21 cells and DF-1 cells infected with the purified TDPV, indicated that the isolation and purification of TDPV was successful, and it can be used for subsequent whole genome sequencing.

2. Comment:

Suitable evidence of the pathology in the bird is presented in figure 1. However, the cytopathic impact shown in figure 2 appears marginal in panel B. More cells appear to be rounded, but this figure does not clear evidence of plaques. Could another field of view be considered or clear labelling of the differences?

Response:

Thanks for your constructive suggestion. To better show the cytopathic impact of this TDPV on its host cells, the experiment of TDPV infection of BHK-21 cells were repeated again including another host cell type DF-1 cells that were also applied to the cell infection experiment. After the optimal of the infection condition, more obvious and typical CPE was observed both in BHK-21 cells and DF-1 cells. In detail, the infected cells exhibited cell rounding, severe aggregation and massive detachment with empty plaques at 96h post-infection. Meanwhile, the progeny viruses were confirmed by RT-PCR test on the supernatants from lysis of the infected cells.

All the new evidences were shown in the revised version. please check it on line 264 to 270. For your convenience, the below images were included in this response letter.

FIG 2 Typical CPE was observed in BHK-21 cells and DF-1 cells induced by TDPV at 96h post-infection. Control group: Uninoculated BHK-21 cells (A) and DF-1 cells (C). Experiment group: BHK-21 cells (B) and DF-1 cells (D) infected with the homogenized supernatants of CAMs with pock lesions.

3. Comment:

Electron microscopy was undertaken, and a virus-like structure identified, however this is not of high resolution and could be omitted (Figure 3).

Response:

We are thankful to the comment about the TEM image of TDPV that the reviewers pointed. We repeated the TEM of the TDPV from infected CAMs again, but the images of the virions were better than that we presented. Considering the earlier suggestion made by the handling editor where it has been pointed out that the pathology, cell biology, genome sequencing and bioinformatic analyses of whole genome of TDPV stand on their own, so the TEM images could be omitted, and therefore, we decided to accept your opinion and delete the part about TEM images of TDPV particles in the new revised version.

4. Comment:

The genome of the virus was determined through genome sequencing. Analysis of the differences with another pigeon virus revealed that this virus contains 44 unique open reading frames (Figure 4).

Response:

Thanks for the reviewers' comments. In this study, the complete genome of this TDPV was sequenced using NGS, and this is the first report of avipoxvirus complete genome sequence from the oriental turtle dove. Then the structure of the TDPV genome, including genome size, A+T content and number of ORFs, was remarkably consistent with other known avipoxviruses in the GenBank database. However, the TDPV genome sequence is significantly different from any other known avipoxviruses, the highest sequence similarity between TDPV and other avipoxviruses (FeP2) was only 92.5%, and the TDPV genome was missing five genes at the corresponding positions compared to the most similar FeP2. Furthermore, the 44 predicted protein-coding genes within its genome cannot be found in any other poxvirus. Within the 44 unique ORFs, only one unique protein coding gene (ORF-011) were predicted to contain a single transmembrane helix, and the biological function of 44 unique ORFs need to be further verified.

5. Comment:

The phylogenetic relationships between the oriental turtle dove pox virus (TDPV) and 63 other chordopoxviruses were undertaken and place this novel virus within the family, suggesting evolutionary relations.

Response:

Thanks to the reviewers for their comments about the phylogenetic relationships of TDPV. The phylogenetic tree constructed using the concatenated amino acid sequences of the ChPV core gene provides evidence that the novel TDPV belonging the subclade A3. Based on the ML tree, we can postulate that the FeP2 and PPV isolated in 2014 and 2023, respectively, are likely to originate from a common ancestor as well as TDPV. The results were further confirmed by the well-supported ML phylogenetic tree constructed using the nucleotide sequences of the P4b gene and DNA polymerase gene of the avipoxvirus genome, which provides evidence that TDPV was most closely related to FeP2, PPV, PEPV, FGPV and CPPV, suggesting that these avipoxviruses likely originated from a common ancestor. Overall, these phylogenetic trees demonstrated the evolutionary genetic linkage of TDPV in

this study, and the TDPV likely should be considered a separate species. This paper enhanced our understanding of the species of avipoxvirus infecting family *Columbidae*, which also contributed to tracking the evolutionary linkage of avipoxvirus infecting this species.

August 8, 2023

Dr. Lei He
Henan University of Science and Technology
Luoyang
China

Re: Spectrum01193-23R3 (A Novel Pathogenic Avipoxvirus Infecting Oriental Turtle Dove (*Streptopelia orientalis*) in China Shows a High Genomic and Evolutionary Proximity with the Pigeon Avipoxviruses Isolated Globally)

Dear Dr. Lei He:

Your manuscript has been accepted, and I am forwarding it to the ASM Journals Department for publication. You will be notified when your proofs are ready to be viewed.

Sincerely,

Wen Chang
Editor, Microbiology Spectrum
